

# Holocene phototrophic community and anoxia dynamics in meromictic Lake Jaczno (NE Poland) using high-resolution hyperspectral imaging and HPLC data.

Stamatina Makri[1], Andrea Lami[2], Luyao Tu[1], Wojciech Tylmann[3], Hendrik Vogel[4,] Martin Grosjean[1]

[1]Institute of Geography & Oeschger Centre for Climate Change Research, University of Bern, Hallerstrasse 12, 3012 Bern, Switzerland

[2]ISE-CNR Institute of Ecosystem Study, 50 Largo Tonolli, 28922 Verbania Pallanza, Italy

[3]Faculty of Oceanography and Geography, University of Gdansk, Bazynskiego 4, PL-80952 Gdansk, Poland

[4]Institute of Geological Sciences & Oeschger Centre for Climate Change Research, University of Bern, 3012, Bern, Switzerland

*Correspondence to*: Stamatina Makri (stamatina.makri@giub.unibe.ch)

**Abstract.** Global spread of hypoxia and altered mixing regimes in freshwater systems is a growing major environmental concern. Climate change and human impact are expected to increasingly deteriorate aquatic ecosystems. The study of processes and drivers of such changes in the past provides a great asset for prevention and remediation in the future. We used a multi-proxy approach combining high-resolution Hyperspectral Imaging (HSI) pigment data, with specific HPLC chlorophylls and carotenoids to examine Holocene trophic state changes and anoxia evolution in meromictic Lake Jaczno, NE Poland. A redundancy analysis RDA including pollen-inferred vegetation cover, temperature and human impacts provides insight into specific conditions and drivers of changing trophic and redox states in the lake. Anoxic and sulfidic conditions established in Lake Jaczno after initial basin infilling 9500 years ago. Until 6700 cal BP, lake trophy was relatively low, water turbidity was high, and green sulfur bacteria (GSB) were abundant within the phototrophic community, suggesting a deep oxic–anoxic boundary and weak stratification. The period between 6700–500 cal BP is characterized by constantly increasing lake production and a gradual shift from GSB to purple sulfur bacteria (PSB), suggesting a shallower oxic–anoxic boundary and pronounced stratification. Yet, the presence of spheroidene and speroidenone in the sediments indicates intermittent anoxia. After 500 cal BP, increasing human impact, deforestation and intensive agriculture promoted lake eutrophication, with a shift to PSB dominance and establishment of permanent anoxia and meromixis. Our study unambiguously documents the legacy of human impact on processes determining eutrophication and anoxia.

**Keywords:** Paleolimnology, Anoxia, Meromixis, Varved sediments, North–East Europe, Holocene, Sedimentary pigments, Human impact

## 1. Introduction

Eutrophication and subsequent oxygen depletion have become primary water quality issues for most freshwater and coastal marine ecosystems globally (Schindler, 2006; Jenny et al., 2016a). Rising mean global temperature can potentially worsen lake anoxia by enhancing water stratification and algal blooms (Adrian et al., 2009;





Woolway and Merchant, 2019). Higher lake trophy and reducing conditions in anoxic bottom waters can have
diverse and profound negative effects on lake ecosystems, such as toxic algal blooms, fish kills, biodiversity loss
(Smol, 2010; Battarbee and Bennion, 2012; Makri et al., 2019), and nutrient recycling into the water column from
the sediment through redox processes (Gächter, 1987; Tu et al., 2019). Hence, the global spread of hypoxia grows
into a major environmental concern.
Temporal and spatial extents of hypoxia/anoxia are influenced by both biological (aquatic production, organic
matter decomposition) and physical (water stratification and lake mixing) factors (Smith and Schindler, 2009;
Friedrich et al., 2014; Jenny et al., 2016b). Environmental and climatic effects such as temperature, seasonality
and extreme events, catchment vegetation, land use, human impact and nutrient input affect lake production and
oxygen supply in the bottom waters.
Observational data of anoxia and aquatic production cover usually only very short periods, which restricts the
understanding of relevant processes and the knowledge of pre-disturbance conditions. This is most relevant for
lake management or restoration. Although recent anoxia and eutrophication have been very well studied and
understood (Naeher et al., 2013; Friedrich et al., 2014; Jenny et al., 2016a), less is known about the onset, cessation
and specific conditions of these changes in the past due to the lack of effective and easily measurable proxies
(Friedrich et al., 2014; Makri et al., 2020). More specifically, to assess lake trophy and/or bottom water
oxygenation, proxies such as sedimentary pigments (Lami et al., 2000; Leavitt and Hodgson, 2001; Guilizzoni and
Lami, 2002), lipid biomarkers (Naeher et al., 2012), diatom (Bennion and Simpson, 2011) and chironomid records
(Little et al., 2000), stable isotopes (Pearson and Coplen, 1978), and redox sensitive elements such as Fe, Mn, Mo,
V, and U (Naeher et al., 2013; Wirth et al., 2013; Costa et al., 2015), have been extensively used so far.
Nonetheless, on long-term Holocene time-scales, most of these proxy records are typically established at a
centennial resolution at best.
Laminated lake sediments are valuable archives of natural and anthropogenic impacts, providing long-term records
via various biogeochemical proxies. Sedimentary photosynthetic pigment records can be effectively used to infer
both changes in algal composition and lake oxygen conditions (Guilizzoni et al., 1983; Leavitt, 1993; Lami et al.,
2000). Chlorophylls, together with their derivatives, and various carotenoids specific to particular groups of algae
can be used to reconstruct overall primary production and the composition of past photosynthetic communities
(Leavitt and Hodgson, 2001). Pigments such as okenone and isorenieratene, which are specific to phototrophic
sulfur bacteria that live in the anoxic sulfidic zones, are regarded as very good indicators of anoxia (Züllig, 1989;
Guilizzoni and Lami, 2002). HPLC-inferred pigments have a coarser temporal resolution due to laborious sample
preparation and time-consuming HPLC measurements. Scanning Hyperspectral Imaging (HSI), a novel non-
destructive method to quantitatively infer the abundance of algal (TChl: chlorophylls and their derivatives) and
bacterial pigments (Bphe: bacteriopheophytins *a* and *b*), offers insight into past trophic and oxygen conditions at
unprecedented µm-scale (sub-seasonal) resolution (Butz et al., 2015; Makri et al., 2020), but with lower speciation
sensitivity.
In this study, we use the varved sediment record of Lake Jaczno (NE Poland) to explore the specific conditions
and mechanisms of trophic and oxygen state changes in the Holocene, under changing climatic and environmental



conditions. Our research has been guided by the following questions: i) Which conditions drove algae dynamics
and oxygen state changes in the Holocene before any significant human intervention? ii) How did climate,
catchment vegetation and erosional input affected the phototrophic community in the lake and iii) How does the
current trophic state and mixing regime of the lake compare with the past? For this, we combined a high-resolution
HSI-inferred record of TChl and Bphe, X-ray fluorescence (XRF) elemental data, and an HPLC pigment record at
coarser resolution but specific in the analysis. Our dataset was compared with vegetation and temperature
reconstruction data to investigate the environmental conditions at times of aquatic primary production and bottom
water oxygenation changes. Lake Jaczno provides ideal conditions to answer these questions. It contains an entirely
varved Holocene sediment record, which has so far only been analyzed for the last 1700 years for productivity,
anoxia (Butz et al., 2016, 2017) and historical land use (Poraj-Górska et al., 2017). Pollen records have revealed
that human pressure was low until the 17[th] century, when landscapes opened and agriculture intensified (Marcisz
et al., 2020). This is rare in Europe. Therefore, this site provides a unique opportunity for a long-term Holocene
assessment of the natural causes and dynamics of meromixis and hypoxia with limited anthropogenic impact until
historic times.
**2.      Study site**
Lake Jaczno (54°16'25.5" N 22°52'15.9" E, 163 m a.s.l, Fig. 1a) is a small, 26 m deep, exoreic, kettle-hole lake
formed sometime after the Weichselian deglaciation ca. 15 ka BP in the Suwałki Lakeland in NE Poland
(Krzywicki, 2002). Lake Jaczno has a total surface area of 0.41 km$^2$ separated in five distinct basins with narrow
sills. It is fed by three permanent inflows (N and W) and one outflow in the south (Fig. 1b). Jaczno is classified as
dimictic and mesotrophic (Tylmann et al., 2013), with incomplete mixing or possibly even meromixis during some
years (Butz et al., 2016). Butz at al. (Butz et al., 2017) found that anoxic, and even meromictic conditions,
established naturally for most of the past 1700 years. Meromixis was interrupted repeatedly following sediment
slumping or flood events.
Microscopic and geochemical analyses in the sediments of Lake Jaczno have revealed seasonal layers (calcareous
biogenic varves) with a complex succession of diatoms and calcite, detrital siliciclastic material (quartz, clays),
organic fragments, and finally amorphous organic matter (Tylmann et al., 2013; Butz et al., 2016; Poraj-Górska et
al., 2017). The lake is surrounded by steep slopes and gullies with ephemeral or perennial water flow, transporting
detrital material to the lake (Fig. 1c).
The catchment area (ca. 9 km$^2$) is covered by glacial tills, sands and fluvioglacial deposits. Modern soils are
classified as cambisols and podsols in the northern part and ferralic cambisols in the southern part of the catchment.
Agricultural lands dominate in the central and northern parts and forests in the southern parts (Fig. 1c). The lake
is surrounded by peatlands and forests dominated by birch, alder and spruce (Weisbrodt et al., 2017). The climate
of the region is continental with a mean annual temperature of 6.8°C and a mean annual precipitation of 600 mm
(Anon, 2017). The lakes in the area typically remain ice covered from December to March (Amann et al., 2014).
Archeological investigations in the Suwałki region indicate sparse or only seasonal human occupation during the
Mesolithic and Neolithic (10,000–3800 cal BP) (Engel and Sobczak, 2012). Around 2000 cal BP human presence
increases in the region with stronghold settlements, animal husbandry and fishing (Kinder et al., 2019). Yet, the
area around Lake Jaczno remained isolated from human influences (Marcisz et al., 2020). Pollen and charcoal





data, and increased soil erosion indicate extensive forest clearance, forest fires and intensified agriculture,
suggesting permanent settlements and higher human impact since 500 cal BP, especially after 150 cal BP (1800
CE) (Kinder et al., 2019; Marcisz et al., 2020). The 1970s are marked by a regeneration of forest cover and a
significant increase of fertilizer use in agriculture (Poraj-Górska et al., 2017; Kinder et al., 2019), which markedly
increased lake primary production (Butz et al., 2016; Poraj-Górska et al., 2017).

### 3.  Materials and methods

Two parallel cores ca. 12.5 m long were retrieved in September 2017, using a UWITEC piston corer. The coring
site was located at the deepest part (24 m water depth) of the lake in the southern basin, which is protected from
direct external inputs (Fig. 1b). The cores were split lengthwise and then described following Schnurrenberger et
al. (2003) and the Munsell color chart (Munsell Color (Firm), 2010). Flood deposits and slumps were identified
based on grain size, mineral content, and sediment structure. First, the core halves were analysed using non-
destructive methods and further analytical measurements were performed after subsampling. The sampling interval
for LOI, CNS and dry-bulk density analysis was 10 cm (ca. 80-year resolution, discrete sampling). For the HPLC
and spectrophotometer analysis, 46 discrete samples (1–2 cm$^3$) were taken every ca. 30–35 cm (ca. 230 years
resolution) taking into account the HSI scanning data and optimization for the proxy-to-proxy calibration of the
HSI indices with spectrophotometer data (Butz et al., 2015). The top 10 cm (last ca. 50 years) were subsampled
continuously every 1 cm.
The chronology is based on 18 radiocarbon AMS dates on taxonomically identified terrestrial plant macrofossils
(Table 1) measured at the Laboratory for Radiocarbon Analysis at the University of Bern. Samples with < 300 μg
C were measured using the gas-source input of the MIni CArbon DAting System (Szidat et al., 2014; Zander et
al., 2020). The age–depth model was calculated using Bacon (rbacon v. 2.4.2; Blaauw et al., 2020; Blaauw and
Christeny, 2011) and the IntCal13 calibration curve (Reimer et al., 2013). Event layers (>3 cm) and slumps were
excluded from the age calculation (Fig. 2). According to changes in lithology, we used model parameters that
allowed for a higher sedimentation rate in the lowermost 137 cm (Fig. 2).
XRF scanning was performed at continuous 2 mm steps using an ITRAX μXRF core scanner (exposure time 20 s,
30 kV and 50 mA) equipped with a Cr-tube at the University of Bern. The results are given as counts (peak area).
From the detected elements, Ti was used as proxy for erosional input from the catchment, Ca as a proxy for
endogenic calcium carbonates, Si/Ti as a proxy for biogenic silica, S, Fe, Mn, and Mn/Fe as proxies of changing
redox conditions (Koinig et al., 2003; Croudace and Rothwell, 2015).
Hyperspectral imaging scanning (HSI) was performed on the freshly oxidized core halves using a Specim PFD-
CL-65-V10E camera (400 to 1000 nm spectral range; 2.8 nm spectral resolution). We used a spatial resolution of
~68 μm per pixel with a spectral sampling of 1.57 nm. Data were processed using the ENVI software version 5.4
(Exelis Visual Information Solutions, Boulder, Colorado) following Butz et al. (2015). The relative absorption
band depths (RABDs) were calculated based on spectral endmembers analysis in ENVI. The RABD$_{673}$ (spectral
region 590–730 nm) was used to detect chlorophylls and their diagenetic products (TChl) and served as a proxy
for aquatic primary production (Leavitt and Hodgson, 2001). The RABD$_{845}$ (spectral region 790–895 nm) was
used to detect bacteriopheophytin $a$ and $b$ ($Bphe$) (Butz et al., 2015, 2016), which is a proxy for anoxia and



meromixis as described in Makri et al. (2020). Bphe *a* and *b* is produced by anoxygenic phototrophic purple sulfur
and non-sulfur bacteria that proliferate in illuminated anoxic habitats (Yurkov and Beatty, 1998; Madigan and
Jung, 2009). Green sulfur bacteria produce bacteriochlorophyll *c*, *d* and *e*, which do not absorb in the $RABD_{845}$
range. Therefore, HSI-inferred Bphe reflects purple bacteria abundance.
The spectral indices were calibrated with absolute pigment concentrations of 46 selected sediment samples (1 cm$^3$)
measured by spectrophotometry (Shimadzu UV-1800). Pigments were extracted using pure acetone. The
supernatant was evaporated under nitrogen, and extracts were subsequently redissolved in 2 ml of pure acetone
(method adapted from Schneider at al. 2018). For the calculation of Bphe concentrations, we used the molar
extinction coefficient for Bphe *a* by Fiedor et al. (2002). For TChl, we applied the molar extinction coefficient for
chlorophylls and chlorophyll derivatives by Jeffrey et al. (1975). The performance of the proxy-to-proxy linear
regression models was assessed using the coefficient of determination ($R^2$) and the root mean square error of
prediction (RMSEP) (Butz et al., 2015) run in R (R Core Team, 2015). The calibration model for TChl showed an
$R^2$ of 0.91 ($p < 0.001$) and a RMSEP ~8 % (Fig. S1a). The calibration model for Bphe showed an $R^2$ of 0.95 ($p <$
0.001) and a RMSEP ~6 % (Fig. S1b). The Shapiro–Wilk and the Kolmogorov–Smirnov tests of the residuals
show that they are most likely normally distributed, suggesting that inferences can be made with both models.
HPLC analysis was conducted on the same 46 samples used for the proxy-to-proxy calibration. Chlorophyll,
chlorophyll derivatives and carotenoids were measured using ion pairing reverse-phase (Mantoura and Llewellyn,
1983; Hurley, 1988). The system used a UV-VIS detector set at 460 nm and 656 nm for carotenoids and
chloropigments, respectively. The results were corrected for water content and expressed as nmol g OM$^{-1}$ (Züllig,
1982; Guilizzoni et al., 1983; Lami et al., 1994). According to Jeffrey et al. (2011), Guilizzoni and Lami (2002),
chlorophyll *a*, *ββ*-carotene, pheophytin *a*, and pheophytin *b* are considered as indicators of total algal biomass.
Chlorophyll *b* and lutein are associated with green algae. *B*-carotene, dinoxanthin (pyrophytes), diadinoxanthin
(siliceous algae), fucoxanthin (diatoms), diatoxanthin (chrysophytes) and alloxanthin (cryptophytes) are related to
brown algae. Echinenone and zeaxanthin are associated to blue–green algae, and myxoxanthophyll and
canthaxanthin to colonial and filamentous cyanobacteria (Leavitt and Hodgson, 2001). K-myxol (4-keto-myxol-
2'-methylpentoside) is associated with N-fixing cyanobacteria (*Anabaena flos-aquae*) (Kosourov et al., 2016).
Pheophorbide *a* is considered as indicator of grazing. In the phototrophic bacteria community, BChl *a* is common
to all anoxygenic phototrophic purple bacteria. Okenone (*Chromatium sp.*) is associated with purple sulfur bacteria
(PSB), whereas spheroidene and spheroidenone (*Rhodopseudomonas sphaeroides*) are related to purple nonsulfur
bacteria (PnSB). Both groups are able to oxidize sulfide. Yet, PSB store any S$^0$ formed intracellularly, whereas
PnSB do so outside the cell (Madigan and Jung, 2009). The main difference between the two groups is that PSB
are strong photoautotrophs, whereas PnSB are physiologically versatile and can grow well both phototrophically
and in darkness via fermentation or anaerobic respiration (Madigan and Jung, 2009). *R. sphaeroides* are also
excellent N-fixing bacteria. Oxygen tolerance varies among species, with *R. sphaeroides* being able to grow under
vigorous aeration. Spheroidenone is produced by *R. sphaeroides* only when even small amounts of oxygen are
present (Züllig, 1989). Hence, the presence of spheroidenone is used as an indication of better oxygen conditions,
whereas the presence of spheroidene with parallel absence of spheroidenone is used as an indication of meromictic
conditions (Züllig, 1989; Guilizzoni and Lami, 2002). Isorenieratene is associated with GSB (*Chlorobium sp.*).
GSB have low light requirements and can cope with low light availability, occupying deeper layers in stratified





lakes (Montesinos et al., 1983). Hence, a dominance of GSB over PSB is used as an indicator of a deeper oxic–
anoxic boundary (Montesinos et al., 1983; Itoh et al., 2003).

Total organic carbon (TOC) was determined by Loss on Ignition (LOI; Heiri et al., 2001). Total carbon (TC) and
total nitrogen (TN) were measured with a CNS-Analyzer (Elementar vario EL cube). Total inorganic carbon (TIC)
was calculated by the difference between TC and TOC (Enters et al., 2010). The TOC/TN ratio was used to infer
changes in OM sources (Meyers, 2003). The lithogenic flux was calculated based on the residual calculation after
removing the organic matter and carbonate fraction by LOI.

Statistical analysis was performed in R (R Core Team, 2015). To define the sedimentary lithotypes we performed
an hierarchical unconstrained clustering on the geochemical proxies (XRF data: Ti, Ca, Si/Ti, Si, S, Fe, Mn, Mn/Fe;
HSI: TChl, Bphe; TOC, TIC, TN, TOC/TN, Fig. 3) using the Euclidean distance matrix and the ward.D2 clustering
method (Murtagh and Legendre, 2014). On the same dataset, we performed a PCA analysis with the samples
grouped based on the unconstrained clustering to investigate the relationships between the lithotypes and the
geochemical variables (Fig. S2). The data were log transformed and scaled before statistical analysis. We
performed a redundancy analysis (RDA) using the Vegan package (Oksanen et al., 2016) in R to relate the pigment
matrix i.e HPLC- and HSI-inferred pigment concentrations (Hellinger-transformed variables) to the environmental
variables i.e. temperature (Heikkilä and Seppä, 2010), arboreal pollen (AP), non-arboreal pollen (NAP) (Kinder et
al., 2019; Marcisz et al., 2020) and lithogenic flux (log transformed variables). The elements were plotted using
scaling 2 (see Borcard et al., 2011, pp. 166–167). This analysis was followed by a permutation test in R to test for
significance in the redundancy analysis (Legendre and Legendre, 1998; Borcard et al., 2011). The zones of pigment
data were defined by constrained clustering using the Bray distance and ward.D2 linkage method in R.
**4.**      **Results and interpretation**
**4.1**      **Chronology**
The age–depth model (Fig. 2) reveals a basal age of ca. 9500 cal BP. The model shows a stationary distribution,
matching prior and posterior accumulation rates, and a smooth sediment accumulation as indicated by its memory
or variability (Fig. 2). Three radiocarbon samples (Fig. 2, in red) have calibrated ages that do not fit with the 95 %
confidence interval. Based on the lithology and the much older ages, these samples were considered as containing
reworked carbon and were excluded from the Bacon model. The sediment sequence is entirely laminated
throughout the Holocene showing regular continuous sedimentation without any hiatus. In the lowermost section
(1257–1120 cm), sedimentation rates (SR) are relatively high (0.5 cm y$^{-1}$) and the mean age error (95% confidence
interval) is ca. ±160 years. Numerous event layers and slumps characterize the part between 1120 cm and 800 cm
where the SR is ca. 0.2 cm y$^{-1}$ and the mean age error is ca. ±200 years. From 800 cm to the top, the sediment is
continuously varved and the SR is 0.1 cm y$^{-1}$ (0.2 cm y$^{-1}$ in the last 500 years). The mean age error in this section
is ca. ±140 years.
**4.2**      **Lithotypes and biogeochemical proxies**
Figure 3 shows the biogeochemical data that defined four sedimentary lithotypes A–D. Fig. S3 (supplementary
material) shows the RGB images and the biogeochemical composition of selected close-ups within the sediment



sequence. Lithotype A and B appear in segments between ca. 9500–6800 cal BP (Fig. 3). Lithotype A, at the
bottom part (9500–9200 cal BP), consists of light greenish grey (GLEY 2 7/2) fine sand and continues with pale
yellow (2.5Y 7/3) and grey (2.5Y 5/1) laminations with light greenish grey silty lenticular bedding. This part is
characterized by high detrital inputs (Ti, lithogenic flux), moderate carbonate content (Ca, TIC), low production
and biogenic silica (HSI-TChl, Si/Ti) and low TOC. Low HSI-Bphe and S, and higher Mn/Fe ratio indicate
effective oxygenation of bottom waters. From ca. 9200–8500 cal BP, lithotype B is introduced and is characterized
by slightly higher production (HSI-TChl) and biogenic silica (Si/Ti); carbonates (TIC, Ca), TOC and TN contents
increase, whereas higher S, Fe and HSI-Bphe, and lower Mn/Fe ratio indicate the development of anoxic (sulfidic)
conditions. Between ca. 8500–6800 cal BP, lithotype A continues with varved sediments; starting with biogenic
pale yellow (2.5Y 7/3) and grey (2.5Y 5/1) varves with intercalated reddish brown (2.5YR 5/4) and reddish black
(2.5YR 2.5/1) laminations rich in clastic material and iron oxides. In the second half of this part, varves are less
well-preserved with several intercalated clastic-rich laminations. Based on color, layer thickness, and grain size
we interpret these intercalated layers as event (flood) deposits. In this period, lithotype A is characterized by high
detrital input (Ti, lithogenic flux); primary production (HSI-TChl) remains unchanged and carbonates (Ca, TIC)
show increased variability. Biogenic silica (Si/Ti), TOC and Fe slightly decrease. S decreases, HSI-Bphe is very
low or absent and Mn/Fe increases, suggesting better oxygen conditions.

Lithotype C occurs between ca. 6800–500 cal BP and consists mainly of light grey (2.5YR) and dark grey (2.5YR
4/1) fine biogenic varves, with some dispersed event layers that occur only at the beginning of this period until ca.
6000 cal BP. This period is characterized by low erosional input (Ti, lithogenic flux), gradually increasing
production (HSI-TChl) and TOC content, fluctuating biogenic silica (Si/Ti) and constantly high carbonates content
(Ca, TIC). S counts are minimal. HSI-Bphe is mostly present suggesting the development of anoxic conditions in
the hypolimnion. Mn/Fe seems to fluctuate, with higher values when HSI-Bphe is lower and vice versa.

Lithotype D occurs from ca. 500 cal BP to the present and consists of biogenic pale yellow (2.5Y 7/3), grey (2.5Y
5/1) and dark grey (2.5Y 4/1) calcareous biogenic varves. This period is characterized by instances of higher
detrital input (Ti) and several intercalated event (flood) layers. Mn counts also increase. Primary production (HSI-
TChl), TOC and TN reach maximum levels, whereas biogenic silica and carbonates (Ca, TIC) decrease. HSI-Bphe
reach maximum values at the top suggesting persistent anoxia in this part. The Mn/Fe ratio and HSI-Bphe show
opposite fluctuations indicating phases of better oxygen conditions when Bphe is absent.
**4.3    HPLC pigment stratigraphy**
Figure 4 presents the pigment dataset of individual chlorophylls and carotenoids measured by HPLC in the
Holocene (Fig. 4a), and for the last 50 years (Fig. 4b). The pigments are grouped according to their taxonomic
relation and the zones are defined by constrained clustering, which yielded boundaries that are similar to those of
the sediment lithotypes (Fig. 3).

In zone I (ca. 9500–9200 cal BP), pigment concentrations are very low. Chromophytes are more abundant than
green algae, especially cryptophytes (alloxanthin) and chrysophytes (fucoxanthin). Blue–green algae (echinenone,
zeaxanthin) are present in low concentrations. Grazing (pheophorbide *a*) is low. In the purple bacteria group,





*Chromatium* species (okenone, PSB) are absent, whereas *R. sphaeroides* (spheroidene and spheroidenone, PnSB)
are both present in low concentrations. *Chlorobium sp.* (isorenieratene, GSB) are present in traces.

In zone II (ca. 9200–6700 cal BP), pigment concentrations increase overall. Green algae (chlorophyll *b*, lutein)
still have low concentrations, whereas chromophytes (*β*-carotene) become more abundant especially pyrophytes
(dinoxanthin) and chrysophytes (diatoxanthin) that show a distinctive local maximum around 7300 cal BP.
Colonial filamentous cyanobacteria (canthaxanthin) appear in this zone. Grazing (pheophorbide *a*) starts
increasing around 8300 cal BP. *Chromatium sp.* (okenone, PSB) is mostly absent. *R. sphaeroides* (spheroidene
and spheroidenone, PnSB) have moderate concentrations, whereas *Chlorobium sp.* (isorenieratene, GSB) reach a
maximum around 7300 cal BP.

In zone III (6700–500 cal BP), most pigments concentration increase gradually. Green algae (chlorophyll *b*)
increase significantly. Chromophytes remain abundant. Diatoms and other siliceous algae (diadinoxanthin,
fucoxanthin), and cryptophytes (alloxanthin) show a local maximum around 2000 cal BP. Blue–green algae
(echinenone, zeaxanthin) increase gradually. More colonial filamentous cyanobacteria (myxoxanthophyll) appear
around 2300 cal BP and, together with zeaxanthin, reach a maximum around 2000 cal BP. N-fixing cyanobacteria
(k-myxol) appear at ca. 5000 cal BP. *Chromatium sp.* (okenone, PSB) appear in this zone and increase gradually.
*R. sphaeroides* (spheroidene and spheroidenone, PnSB) also show a gradual increase, whereas *Chlorobium sp.*
(isorenieratene, GSB) decrease to minimum concentrations.

Zone IV (500 cal BP to present), is characterized by a further gradual increase of most pigments, reaching
unprecedented maximum concentrations at the top. In more detail, Fig. 4b shows the distribution of pigment
concentrations in the last 50 years. Most pigments reach maximum values around 1997 CE. *Chromatium sp.*
(okenone, PSB) have high concentrations, whereas *R. sphaeroides* are present producing only spheroidene and
almost no spheroidenone. *Chlorobium sp.* (isorenieratene, GSB) show only trace concentrations around 1997 CE.
**4.4    The relationships between land use, temperature, and pigment stratigraphy**
We applied a redundancy analysis (RDA) to examine the response of our HPLC- and HSI-inferred pigment dataset
to land use changes (arboreal pollen: AP, and non-arboreal pollen: NAP; Kinder et al., 2019; Marcisz et al., 2020),
annual mean temperature variability (Heikkilä and Seppä, 2010) and catchment surface processes (lithogenic flux).
Figure 5 shows the RDA ordination output in a triplot with the explanatory variables (in blue) and response (in
red) variables, as well as the samples divided into the four distinct zones defined by constrained clustering (see
Sect. 4.3). The numerical output shows that the first two axis (RDA 1 20.95 % and RDA 2 11.25 %) explain 36 %
of the variation (unadjusted values). The $R^2_{adj}$ for the constrained ordinations suggests that this model explains ca.
29 % of the variation in the data. The permutation test on the unconstrained ordinations indicates that the first two
axis are significant (p<0.001; Table S1) and represent the data adequately.

The RDA triplot (Fig. 5) shows that AP and NAP play an important role in the distribution of the pigment data
along the first axis (RDA 1). Lithogenic flux and temperature drive pigment variability along the second axis
(RDA 2). Lithogenic flux is strongly correlated with siliceous algae (fucoxanthin, diadinoxanthin) and blue–green
algae (echinenone), as well as enhanced aquatic primary production (HSI-TChl) by green algae (lutein) and



cryptophytes (alloxanthin, *β*-carotene). Lithogenic flux is clearly anticorrelated with PSB (okenone, HSI-Bphe)
and Chl *a*. AP is mainly correlated with GSB pigments (isorenieratene) indicating a deeper oxic–anoxic boundary,
and PnSB (spheroidene and spheroidenone) that suggest a more effective oxygenation of the water column. AP is
also correlated with variables indicating the presence of chromophyte (brown) algae, pyrophytes (dinoxanthin),
chrysophytes (diatoxanthin), as well as some blue–green algae (zeaxanthin). Higher lithogenic input and AP drive
pigment variability in zones I and II. Temperature seems to be correlated with higher production of some green
algae (Chl *b*, pheophytin *b*), increased cyanobacteria abundance (*ββ*-carotene, *β*-carotene) and colonial-
filamentous cyanobacteria (myxoxanthophyll, canthaxanthin). Temperature seems to drive pigment variability
mainly in zone III. NAP is correlated with PSB production (BChl *a*, HSI-Bphe and okenone), overall higher
primary production (Chl *a*, pheophytin *a*), higher grazing (pheophorbide *a*), and N-fixing cyanobacteria (k-myxol).
NAP drives pigment variability in zone IV.
**5.    Discussion**
**5.1    Combining sedimentological and biogeochemical data to infer past lake production and bottom**
**water oxygenation**
The 12.5 m long and almost entirely varved sediment record of Lake Jaczno continuously spans the last ca. 9500
cal yr BP (Fig. 2). The chronology is robust and exclusively based on terrestrial macrofossils. The lithology of
Lake Jaczno (Fig. 3) revealed the deposition of frequent event layers between 8500–7000 cal BP, which likely
reflect a regional catchment/climatic signal as similar features have been observed, for the same period, in the
nearby Lake Szurpiły (Kinder et al., 2020). The physical characteristics of the catchment favored the transport of
lithogenic material into the lake (Fig. 3), thereby possibly affecting the density stratification, light availability, and
subsequently the phototrophic community dynamics. A proper assessment of these changes requires high-
resolution data that is impossible to reach using HPLC data alone. Yet, the combination of high-resolution (μm-
scale) calibrated HSI bulk data for TChl and Bphe, combined with scanning XRF and compound specific HPLC
data, provides a unique opportunity for paleoproduction and paleooxygenation reconstructions at sub-seasonal
scale for multi-millennial-long records (Butz et al., 2017; Makri et al., 2020). This approach is directly applicable
to diverse lacustrine (Butz et al., 2017; Schneider et al., 2018; Makri et al., 2020; Sanchini et al., 2020) and
potentially marine environments (Hubas et al., 2011, 2013) with uncertain past redox state changes.
The calibration of the RABD$_{673}$ and RABD$_{845}$ to absolute pigment concentrations of green pigments (chlorophylls
and diagenetic products) and Bphe (*a* and *b*) respectively, revealed robust calibration statistics (Fig. S1,
supplementary material) with very low uncertainties (ca. 6–8 %) comparable to other studies (Butz et al., 2017;
Schneider et al., 2018; Makri et al., 2020; Sanchini et al., 2020). Between ca. 9200 and 7000 cal BP, the calibration
model of the RABD$_{673}$ for green pigments calculates negative concentrations (Fig. 3). This offset can be produced
by matrix effects, i.e. the variability of the reflectance of the sediment matrix or substances that absorb in the same
range as chlorophylls and their diagenetic products (590–730 nm) (Makri et al., 2020). Interestingly, GSB
(isorenieratene) peak between 9200 and 7000 cal BP (Fig. 4). GSB contain bacteriochlorophyll *c*, *d*, and *e* that
absorb in the same range as chlorophylls and chlorophyll derivatives (Oren, 2011). This could indicate that a part
of the RABD$_{673}$ calibration error may be due to the increased GSB abundance. Nonetheless, the calibration
statistics reveal an overall error of less than 8 %.



### 5.2 Holocene production dynamics and chemocline evolution

The presence of anoxygenic sulfur bacteria throughout our record, combined with chlorophylls, carotenoids and
geochemical evidence, suggests that euxinic conditions prevailed in Lake Jaczno for most of the past 9500 years.
Nonetheless, the changing composition of photosynthetic sulfur bacteria indicates persisting but variable euxinia.
Figure 6 summarizes the Holocene evolution of the relative abundance of PSB, PnSB and GSB, *Chromatium*
(okenone) and *Clorobium* (isorenieratene), the content of spheroidene and spheroidenone pigments produced by
*R. sphaeroides*, and the high-resolution calibrated HSI-TChl and Bphe, with respect to lithogenic flux, climate
variability (annual mean temperature; Heikkilä and Seppä, 2010) and human impact (land use and vegetation
cover; Kinder et al., 2019; Marcisz et al., 2020).

### 5.2.1 Low trophic levels with a deep oxic–anoxic boundary

In the period from 9500 to 6700 cal BP, which corresponds to pigment zones I and II, the phototrophic bacteria
population is dominated by GSB (Fig. 6). A small percentage of PnSB (*R. sphaeroides*) is present and seems to
produce both spheroidene and spheroidenone during this time. *Chromatium* (okenone, PSB) is almost completely
absent. Considering that *R. sphaeroides* produces spheroidenone only when even small amounts of oxygen is
present (Züllig, 1989) we suggest that, in this period, euxinic conditions were already present but the strength or
extent of anoxia was likely weak. HSI-Bphe that corresponds to purple bacteria is very low. Anoxia is mainly a
function of lake stratification and productivity. HSI-TChl, which indicates total primary production, is still at low
levels (Fig. 6). Indeed, the stratigraphy of individual pigments indicates low to moderate in-lake production, which
mainly consists of chromophyte (brown) algae and some colonial cyanobacteria (canthaxanthin) (Fig. 4), which is
also confirmed in the RDA analysis (Fig. 5). Brown siliceous algae are well adapted and tolerant algae species that
thrive in oligotrophic conditions in symbiosis with other algae species and bacteria (Bird and Kalff, 1986; Wetzel,
2001). Similar observations of algae composition were made in Lake Peipsi (Tõnno et al., 2019; Estonia) and Lake
Lazduny (Sanchini et al., 2020; NE Poland).

Temperature gradually increased and a closed forest canopy with pine/birch and later elm/hazel/alder persisted in
the catchment (Fig. 6) (Gałka, 2014). These provided shelter from wind and increased the nutrient pools in the
catchment soils (Bajard et al., 2017). The closed forest canopy, combined with the deep and relatively small basin
(relative depth 3.01 %), favors the establishment of a naturally anoxic hypolimnion (Zolitschka et al., 2015). Yet,
it seems that enhanced permanent stratification was still not established in the lake. This phase of GSB dominance
corresponds to a period of high lithogenic flux or high-energy sedimentation (Fig. 3, 6), as confirmed in the RDA
analysis (Fig. 5). Turbidity currents and underflows can increase turbidity and nutrient availability, but can also
cause sporadic ventilation of bottom waters. Higher suspended matter and/or algal growth would decrease light
availability in the oxic–anoxic boundary. Since GSB and PnSB are more tolerant to low light intensities than PSB
(Biebl and Pfennig, 1978; Parkin and Brock, 1980; Madigan and Jung, 2009), a dominance of GSB and presence
of spheroidene and spheroidenone in the sediments is expected under these conditions. Similar observations were
made in Lake Cadagno (Wirth et al., 2013). GSB often inhabit the lowermost part of stratified water bodies due to
their efficient light capture (Manske et al., 2005; Imhoff, 2014). Higher abundance of GSB could also indicate a
deep oxic–anoxic boundary in the lake (Itoh et al., 2003; Antoniades et al., 2009).





### 5.2.2 Gradually increasing trophy levels with a shallower oxic–anoxic boundary

The period from 6700 to 500 cal BP, which corresponds to pigment zone III, is characterized by a gradual shift in the phototrophic bacterial community to higher PSB abundance, especially after ca. 2000 cal BP (Fig. 6). GSB are present, inhabiting the anoxic layers below PSB and seem to fluctuate as a function of the primary production in the oxic layer (HSI-TChl) and related light availability (Montesinos et al., 1983). When production was higher in the oxic layers, *Chromatium* (okenone, PSB) increased and *Chlorobium* (isorenieratene, GSB) decreased. This is also confirmed by the individual pigment stratigraphy (Fig. 4). Green algae (chlorophyll *b*, lutein) and N-fixing cyanobacteria (k-myxol) increase markedly since ca. 5500 cal BP, indicating a higher lake trophy than before, driven most probably by lake ontogeny and a gradual increase of nutrient availability. The appearance of N-fixing cyanobacteria at that time agrees with this interpretation. Prolonged periods of anoxia leading to intense recycling of phosphorous from the sediments would decrease the N:P ratio in the water column promoting nitrogen fixation by N-fixing algae (Howarth et al., 1999; Vitousek et al., 2002). Similar trends in lake trophy evolution are reported from nearby Lake Szurpiły (Kinder et al., 2019) and Lakes Albano and Peipsi (Lami et al., 2000; Guilizzoni and Lami, 2002; Tõnno et al., 2019). An increase of *Chromatium* (okenone, PSB) over *Chlorobium* (isorenieratene, GSB) with increasing lake trophy has been reported from other lakes, e.g. Lake Albano in Italy (Lami et al., 1994), Little Round Lake in Canada (Brown et al., 1984), and Lake Hamana in Japan (Itoh et al., 2003). *R. sphaeroides* (PnSB) are also present producing both spheroidene and spheroidenone, suggesting phases of effective aeration of bottom waters (Züllig, 1989).

The catchment is continuously densely forested and human impact is very low (Fig. 6). The RDA analysis points to a temperature driven pigment variability in this zone, but mainly for cyanobacteria abundance (Fig. 5). Cyanobacteria can benefit from higher water temperature, yet nutrient inputs have in most cases a much stronger and synergetic effect (Lürling et al., 2018). Temperature variability did not seem to have affected lake stratification directly. However, seasonality, precipitation and windiness play an important role in lake circulation and are not reflected in the annual mean temperature variability. Hence, the role of climate may be underestimated. The oxic–anoxic stratification was enhanced in this period but permanent perennial anoxia was still not established as indicated also by the low HSI-Bphe concentrations. The increase in PSB abundance suggests a shallower oxic–anoxic boundary (Itoh et al., 2003). It appears that during most of the Holocene, anoxia was largely influenced by primary production and lithogenic flux. The case of Lake Jaczno is different from e.g Lake Łazduny (Masurian Lake District, NE Poland; Sanchini et al., 2020), where erosional input is negligible and anoxia was mainly a function of primary production and forest cover.

### 5.2.3 20th century eutrophication, shallow oxic–anoxic boundary and meromixis

In the period from 500 cal BP to the present, which corresponds to pigment zone I, phototrophic sulfur bacteria composition changed to an almost complete dominance of purple bacteria (Fig. 6). Between 500–200 cal BP, HSI-Bphe and the absolute concentrations of PSB (okenone,) and PnSB (spheroidene and spheroidenone) are at a minimum, but dominate the phototrophic bacteria community since GSB are completely absent (Fig. 4, 6). Lake production (HSI-TChl) also decreases while lithogenic flux increases (Fig. 6). These suggest an oxic rather than anoxic phase during this period, with some intervals of weak euxinia. Increased Mn accumulation during this time (Fig. 3) supports the indications of rather oxygenated bottom waters.



Between 200 cal BP to the present, when human impact starts to increase in the catchment (Fig. 5), PSB increase
as well. Presence of spheroidene and only trace concentrations of spheroidenone (PnSB) and isorenieratene (GSB)
suggest increasing and gradually persisting anoxia. Intensive agriculture in the last 100 years and use of fertilizers
increased primary production (HSI-TChl) substantially to unprecedented levels in the 1990s, relative to the
Holocene baseline. This is also reflected in the individual pigment stratigraphy (Fig. 4). Bphe reaches maximum
levels suggesting persisting anoxia and mostly meromictic conditions in the lake, especially since the 1970s when
gradual afforestation in the catchment is observed. This is also supported by the HPLC-inferred composition of
phototrophic bacteria (Fig. 3).
In this period, the high-resolution HSI-Bphe record indicates that the intervals of lowest AP in the catchment
coincide with absence of Bphe, indicating oxic bottom waters. Bphe increases again only when AP and the tree
canopy recovers (Fig. 6), with a parallel absence of spheroidenone (PnSB), suggesting meromictic conditions.
Butz et al. (Butz et al., 2016, 2017) showed that these intervals of low AP and low or absent Bphe in the sediments
were accompanied by strong pulses of terrigenous material from the catchment. The role of human impact with
regard to anoxia and interrelated catchment processes (deforestation/afforestation and nutrient inputs) has also
been shown in other lakes with diverse timing of human impact onset. For example, Lake Moossee (Makri et al.,
2020) and Soppensee (Lotter, 1999) on the Swiss Plateau, Lakes Albano and Nemi in Italy (Guilizzoni et al.,
2002), Lake Zazari in Greece (Gassner et al., 2020) with an early Mid-Holocene human impact, and Lake Szurpiły
(Kinder et al., 2019) in the vicinity of Lake Jaczno with a late human impact, mainly in the last 500 years.

## 6.    Conclusions

In this study, we used a multiproxy approach, combining high-resolution HSI pigment data with lower resolution
HPLC-inferred concentrations of specific algal pigments, and geochemical data to investigate algal community
composition and its relationship with aquatic production and water column oxygenation in a 9500-years sediment
record from NE Poland. Land use changes, vegetation cover and climate variability were also taken into account.
Our aim was to examine factors that determine trophic state changes and lake stratification, in a lake system with
stable catchment vegetation and low human impact until very recent times.

The Holocene sedimentary pigment and geochemical record of Lake Jaczno revealed distinct changes in lake
trophy and stratification states, mainly driven by the catchment evolution, lithogenic flux, nutrient input and
subsequent increase in primary production. The lake had a first phase (9500–6700 cal BP) of low production that
consisted mainly of brown algae in the oxic zone, yet an early immediate establishment of weak euxinic conditions
in a deep water column dominated by GSB in its anoxic zone. Increased suspended loads, turbidity currents and
underflows seem to have increased turbidity and restricted the proliferation of PSB at the deep oxic–anoxic
boundary. Between 6700–500 cal BP, primary production increased gradually with higher contributions of green
algae and cyanobacteria, following lake ontogeny in a continuously densely forested catchment. The oxic–anoxic
boundary became gradually shallower with a shift from GSB to PSB. The composition of phototrophic bacteria
and the presence of spheroidene and spheroidenone (PnSB) in the sediments suggest pronounced yet intermittent
euxinia in the lake. Between 500 cal BP to the present, lake trophy increases dramatically, especially in the last
100 years, due to intensified human impact. Eutrophication accompanied by catchment deforestation and



subsequent afforestation after land abandonment were the main driving forces for the establishment of permanently
anoxic and meromictic conditions in the modern lake.

This study highlights the great potential of calibrated and validated HSI measurements combined with HPLC data.
Lake Jaczno provided a rare site to explore the mechanisms that can potentially induce changes in lake mixing,
production and sustained bottom water anoxia in times from minimum to intensive human impact, in a naturally
stratified lake system. Our findings, together with findings from other lakes across Europe, can greatly expand our
understanding on these major environmental problems while providing a tailored toolset for implementing
effective remediation techniques in the future.
**Data availability**
The data will be made available at PANGAEA
**Author contributions**
**Stamatina Makri**: Investigation, Data Curation, Formal analysis, Writing - Original Draft, Visualization. **Luyao**
**Tu:** Investigation, Writing - Review & Editing. **Andrea Lami:** Investigation, Writing - Review & Editing.
**Wojciech Tylmann:** Writing - Review & Editing. **Hendrik Vogel:** Writing - Review & Editing, **Martin**
**Grosjean:** Conceptualization, Methodology, Writing - Review & Editing, Supervision, Funding acquisition.
**Competing interests**
The authors declare that they have no conflict of interest.
**Acknowledgments**
This study was funded by the Hans Sigrist Stiftung and the Swiss National Science Foundation Grants (SNF
200021_172586). We thank Andre F. Lotter, Willi Tanner, Paul Zander and Maurycy J Żarczyński for their help
during field work. We thank Petra Boltshauser-Kaltenrieder for plant macrofossils identification. Further, we
acknowledge Daniela Fischer and Patrick Neuhaus for their assistance in the lab.

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






**Figure 1: a) Localization of Lake Jaczno. b) Lake bathymetry (modified from Poraj-Górska et al., 2017) and coring**
**position c) Slopes and land use maps of the catchment (modified from Poraj-Górska et al., 2017) d) Seasonal limnological**
**parameters in 2013 CE (Butz et al., 2016).**






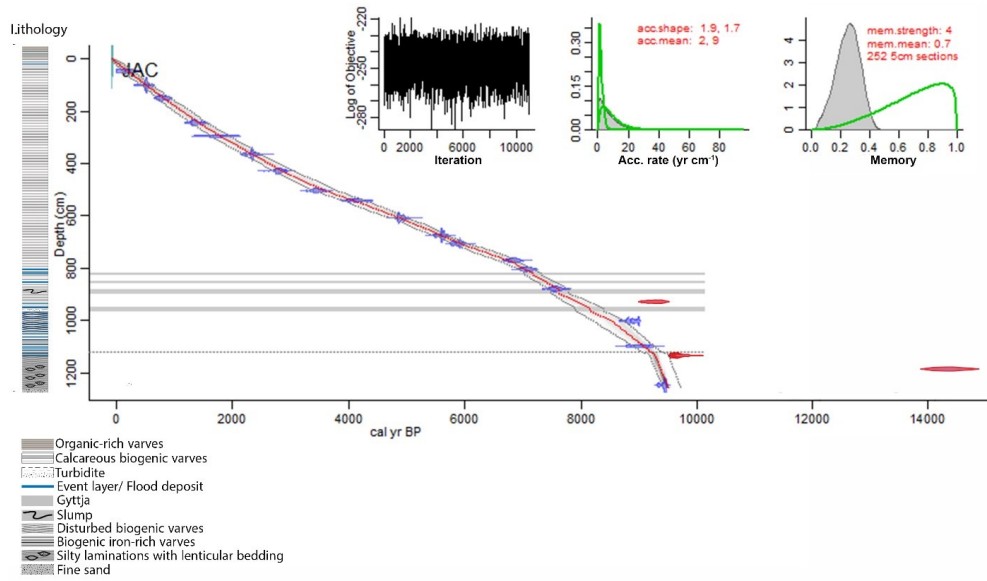



**Figure 2: Age depth model and lithology of Lake Jaczno. The red line is the modelled chronology using**
**Bacon (Blaauw and Christeny, 2011; Blaauw et al., 2020); the excluded outliers are shown in red. The grey**
**dotted lines indicate the 95 % (2σ) probabilities. The grey horizontal areas indicate event layers (>3 cm)**
**excluded from the model. The horizontal dashed line marks the boundary of a higher sedimentation rate**
**(model parameter). The upper left panel shows the log objective vs. MCMC iteration that indicates a**
**stationary distribution. The middle and right panels indicate the distributions (prior in green, posterior in**
**grey) for the accumulation rate and the memory, respectively.**











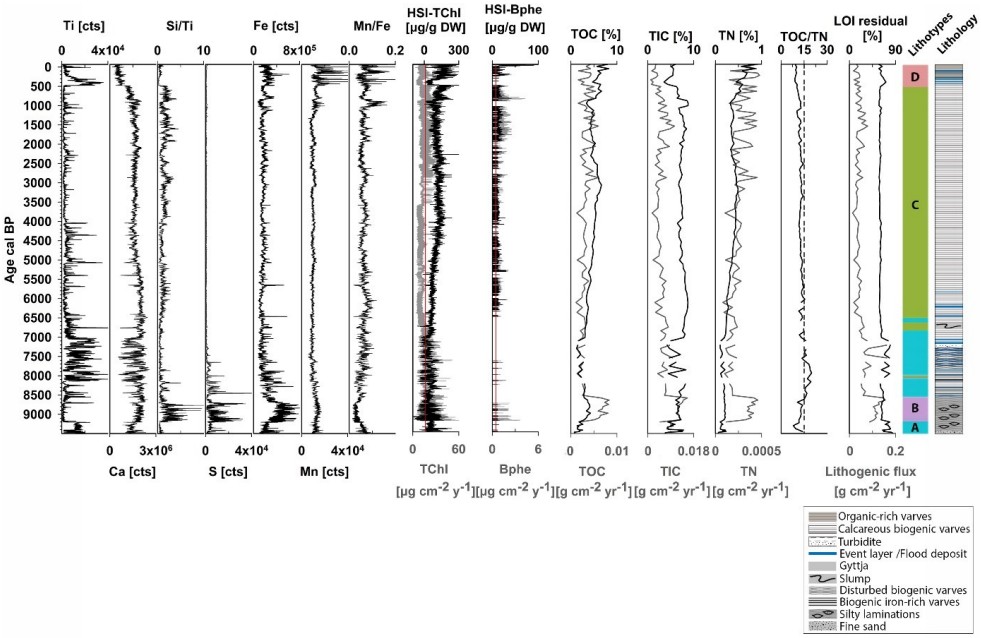

**Figure 3: Selected biogeochemical proxies that defined the four sedimentary lithotypes A–D (in different color) after unconstrained clustering and PCA analysis (Fig. S2). On the right: sediment lithology based on visual examination.**



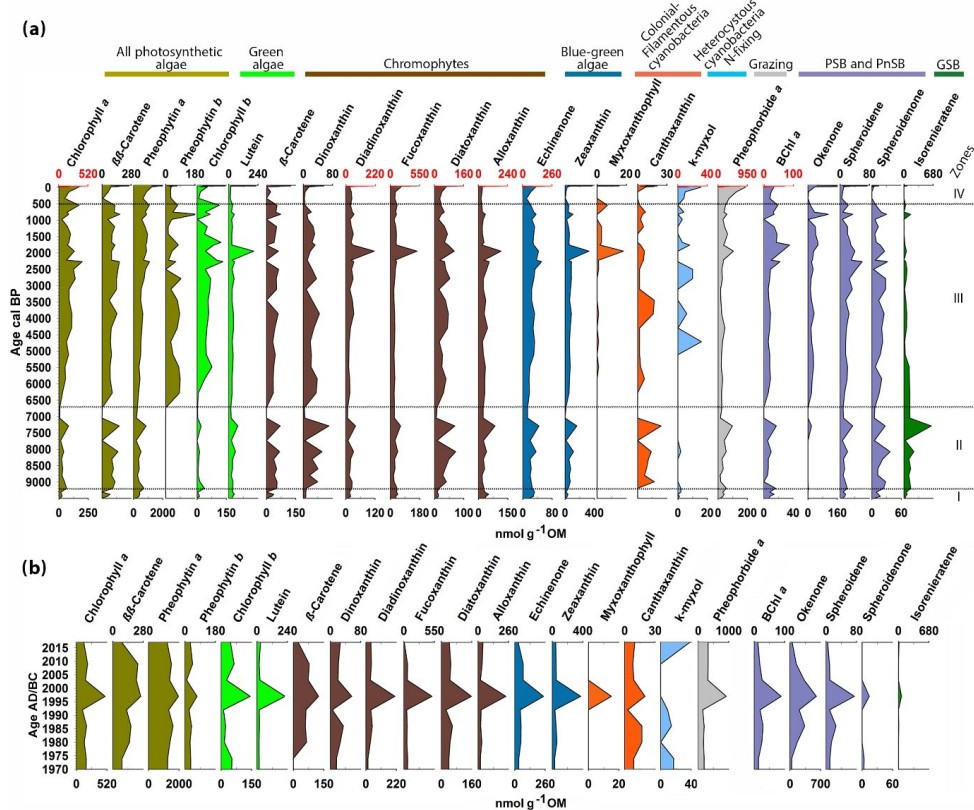


**Figure 4: Chlorophyll, chlorophyll derivatives, carotenoids and bacterial pigments concentrations**
**measured by HPLC a) for the entire Holocene, and b) for the last 50 years. The zones are defined by**
**constrained hierarchical clustering. The different colors indicate different algal groups based on the**
**pigments' taxa affiliation. The occasional red scale on top marks the significantly higher concentrations of**
**these pigments in the last 50 years.**





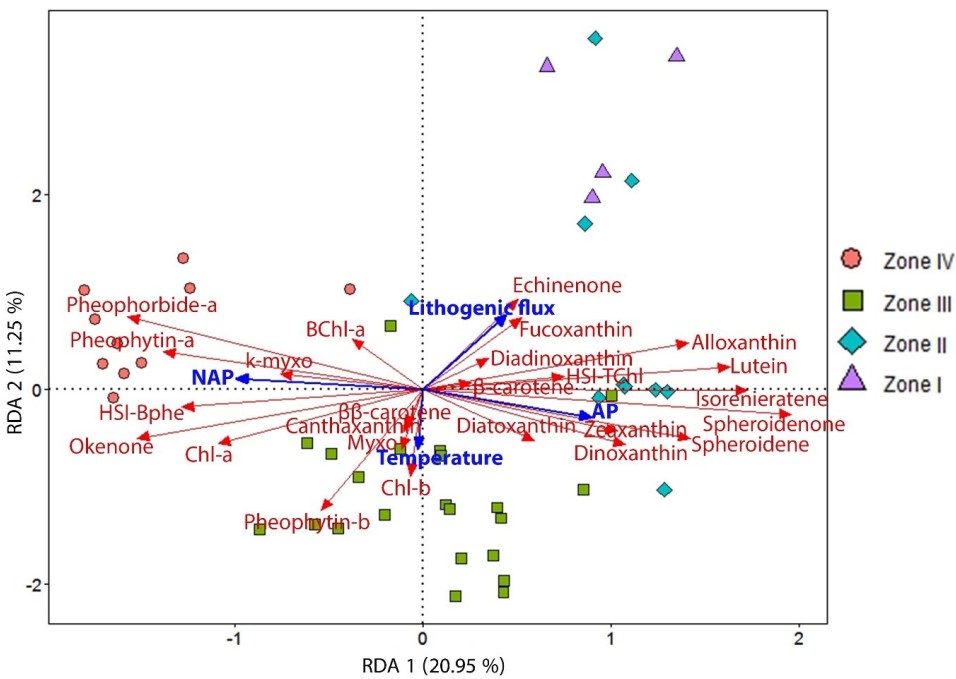


**Fig 5: RDA triplot showing the explanatory variables (AP, NAP, temperature and lithogenic flux) in blue,**

**and the response variables (HPLC- and HSI-inferred pigment concentrations) in red. The samples are**

**grouped according to the pigment zones (Fig. 4a).**






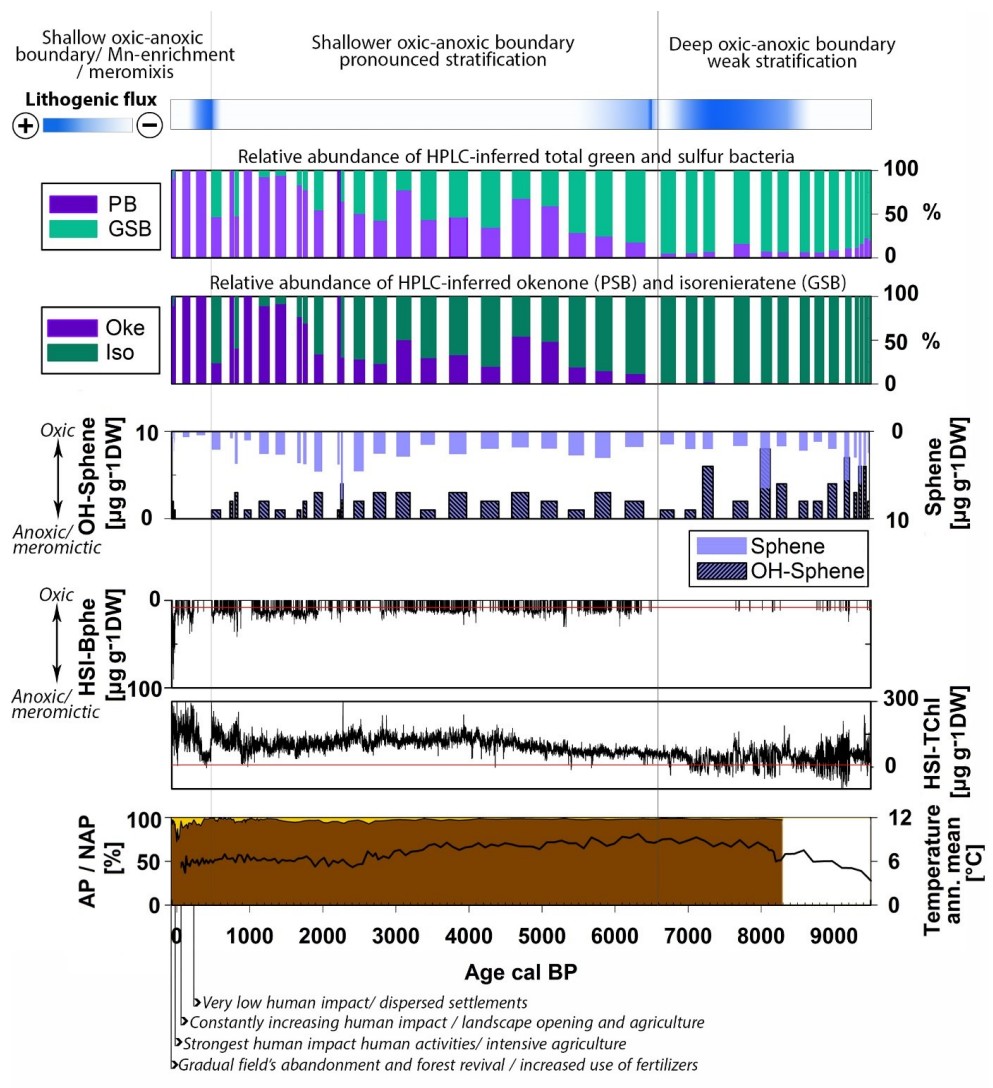

**Figure 6: Holocene summary of the relative abundance of purple bacteria (sum of PSB and PnSB) and GSB,** *Chromatium* **(okenone, PSB) and** *Clorobium* **(isorenieratene, GSB), the content of spheroidene and spheroidenone pigments produced by** *R. sphaeroides* **(PnSB), and the high-resolution calibrated HSI-TChl and HSI-Bphe concentrations. Top: indication of lithogenic flux and general evolution of the chemocline. Bottom: AP/NAP percentages (Kinder et al., 2019; Marcisz et al., 2020) with archaeological evidence of human impact, and the annual mean temperature variability (Heikkilä and Seppä, 2010).**



**Table 1: Radiocarbon age results and calibrated ages. Uncertainties for [14]C ages refer to 68 % probabilities**
**(1σ), whereas ranges of calibrated ages refer to 95 % probabilities (2σ). Outlier samples are marked with**
**an asterisk. Indet: indeterminable, dicot: dicotyledonous.**

| Sample ID | Material | C mass (µg C) | Age [14]C BP | Age (cal BP)[a] | Age range (cal BP)[b] | Graphite/ Gas |
|---|---|---|---|---|---|---|
| BE-10957.1.1 | *Betula alba* fruit, woody scale, *Pinus sylvestris* needle base | 104 | 132±64 | 142 | 0–284 | gas |
| BE-10958.1.1 | *Betula alba* fruit scale, woody scale, dicot leaf fragment | 219 | 482±44 | 755 | 679–904 | graphite |
| BE-10959.1.1 | *Pinus sp.* periderm, coniferous wood and periderm fragment, *Betula alba* fruit fragments, coniferous scales | 118 | 872±55 | 790 | 694–912 | graphite |
| BE-10960.1.1 | *Pinus sp.* periderm*, Betula alba* fruit fragments, conifer scale, Pinus sp. periderm, *Betula alba* fruit fragments | 64 | 1460±67 | 1364 | 1283–1522 | gas |
| BE-10961.1.1 | *Betula alba* fruit fragments, conifer scale, *Betula alba* fruit, semi–charred periderm | 19 | 1781±127 | 1706 | 1410–1987 | gas |
| BE-10962.1.1 | *Alnus glutinosa* fruit, *Betula alba* fruit fragments, needle/leaf indet, male anthere indet | 268 | 2321±39 | 2341 | 2180–2458 | graphite |
| BE-10963.1.1 | *Betula alba* fruit fragments, *Pinus sp.* periderm, male anthere indet, *Betula alba* fruit fragments, *Pinus sp.* periderm, conifer scales | 73 | 2677±69 | 2801 | 2545–2959 | gas |
| BE-10964.1.1 | *Betula alba* fruit fragments, *Pinus sp.* periderm, male anthere indet, *Betula alba* fruit fragments | 74 | 3229±72 | 3458 | 3259–3633 | gas |
| BE-10965.1.1 | *Alnus glutinosa* fruit fragment, *Betula alba* fruit fragments, conifer scales, dicot leaf fragments | 130 | 3758±60 | 4125 | 3927–4383 | graphite |
| BE-10966.1.1 | Male anthere indet, dicot leaf fragments, conifer scales, indet scale, wood indet | 399 | 4322±36 | 4887 | 4836–4972 | graphite |
| BE-10967.1.1 | Dicot leaf fragments, indet scale, male anthere indet | 428 | 4860±37 | 5601 | 5491–5650 | graphite |
| BE-10968.1.1 | Dicot leaf fragments, wood remains | 348 | 5144±39 | 5906 | 5753–5989 | graphite |
| BE-10969.1.1 | Dicot leaf fragments, indet periderm, wood remains | 183 | 5998±57 | 6839 | 6678–6977 | graphite |





| BE-10970.1.1 | Deciduous woody scale | 365 | 6153±41 | 7062 | 6942–7166 | graphite |
|---|---|---|---|---|---|---|
| BE-10971.1.1 | *Betula alba* fruit fragments, *Pinus sp.* periderm, conifer scale, wood indet | 111 | 6699±79 | 7567 | 7439–7674 | graphite |
| *BE-10972.1.1 | *Pinus sylvestris* needle fragments, wood indet | 58 | 8291±99 | 9279 | 9029–9475 | gas |
| BE-10973.1.1 | Dicot leaf fragments, deciduous periderm, wood indet | 996 | 8018±22 | 8896 | 8778–9007 | graphite |
| BE-10974.1.1 | Betula alba fruit fragments, *Pinus sylvestris* needle fragments, dicot leaf fragments | 117 | 8094±91 | 9019 | 8651–9287 | gas |
| *BE-10975.1.1 | *Pinus sylvestris* needle fragments, *Betula alba* fruit fragments, indet periderm | 280 | 8715±54 | 9677 | 9548–9887 | graphite |
| *BE-10976.1.1 | Dicot. leaf fragments, male anthere indet, conifer needle tip, indet periderm | 308 | 12446±69 | 14580 | 14198–14994 | graphite |
| BE-10977.1.1 | Dicot leaf fragments, *Pinus sp.* periderm, woody scale | 999 | 8388±22 | 9440 | 9318–9479 | graphite |

[a] Median probability (Stuiver and Reimer, 1993)
[b] Calibrated age range with the IntCal 13 calibration curve (Stuiver and Reimer, 1993; Reimer et al.,
823  2013)

