# Peer review of "Holocene phototrophic community and anoxia dynamics in 1"

_Biogeosciences, 2020_

## Referee Comment (RC1) · Anonymous Referee #1 · 8 Nov 2020

Manuscript number: bg-2020-362 Title: Holocene phototrophic community and anoxia dynamics in meromictic Lake Jaczno (NE Poland) using high-resolution hyperspectral imaging and HPLC data

Makri et al present a very detailed record of variations in phytoplankton community composition and associated changes in redox conditions of a Polish lake. This lake has already been studied thoroughly in previous publications. However, the authors present new data together with these published records to make a very nice comparison of pigments and trace element records. Very elegant is the combination of high-

resolution techniques to reconstruct short fluctuations in environmental conditions that the lake experienced with traditional techniques that provide high compound specificity, though at the expense of the high temporal resolution. The combination of these techniques provides large insight into the changes in water column conditions and species composition during the lake's history. This manuscript is suitable for Biogeosciences after consideration of mostly minor comments as outlined below.

Title: redox dynamics

Line 15: altered mixing regimes – what does that mean? Is this aspect related to hypoxia or any other reasons? I guess the main problem with changed mixing regime is the change from a well-mixed system to meromixis? Please clarify.

Line 19: change sentence so that you state pigment analysis using two different techniques. While one method enables high spatial resolution pigment analysis (though only raw data), the HPLC data allow high compound specificity. This should be better explained here.

Line 43: The Diaz and Rosenberg papers about Hypoxia would be important references to cite here.

Line 73: Total chlorophylls or only Chlorophyll-a and derivatives considered here? Including Chlorophyll-b and c? Please clarify.

Line 84: Related to my comment above. Please reformulate to low-resolution pigment record using HPLC analysis with high compound specificity, which cannot be achieved by the hyperspectral record.

Line 90: Remove sentence 'This is rare in Europe.' This sentence is not useful.

Line 99: Remove Butz et al. in brackets, because it is noted twice.

Line 157: Are bacteriopheophytin a and b both detected and distinguished by hyperspectral and HPLC techniques? It would be better to separate the records of both

compounds to establish if species-composition changes in the sedimentary record of the lake need to be considered for the reconstructions, because both compounds are not necessarily produced in the same quantities from the same species.

Line 160: Are bacteriochlorophyll c, d and e present as well? If so, are they reconstructed by the HPLC technique? This also shows that the different bacteriochlorophylls and their pheophytins should be distinguished throughout the manuscript instead of using Bphe as abbreviation for the sum of these compounds.

Line 183: blue–green algae are also cyanobacteria. Please distinguish which forms of cyanobacteria can be reconstructed by these two pigments or are these indicators widespread in all cyanobacteria?

Line 186: Pheophorbide a is considered as indicator of grazing – Please add reference to support this. It is a derivative of chlorophyll like other derivatives and can also simply form by degradation/structural alteration, which is not limited to grazing.

Line 233: Unclear why the age uncertainty is high in the varved part of the sedimentary record. These are annual layers, so age determination should be up to a few years only? How to explain this?

Line 331: The chronology is robust and exclusively based on terrestrial macrofossils – Related comment to the previous comment. Why is there no higher precision in the age record of the upper part of the record as it is varved? Other radiometric dating techniques that are useful, such as 210Pb dating? The high uncertainty of about 140 years indicates that the age model appears less robust than it is expected to be due to the presence of varves?

Line 481: The data should be uploaded to PANGAEA now so the link to the datasets can be included into the final version of the paper.

———————————————

---

## Referee Comment (RC2) · Anonymous Referee #2 · 24 Nov 2020

The paper by Makri et al makes use of a high resolution, laminated lake sediment record from Poland, which covers the last 9500 years. The authors use high-resolution (mm-scale) Hyperspectral Imaging pigment data together with low resolution (dm scale) chlorophyll and caretonoids data to document the impact of humans into the lake and nearby environment. The lake is particularly suited for such a study, because pollen evidence document that the region is used by humans only since about 500 years. The region was in a natural state apparently for most of the Holocene. The lithology is presented as three main units, which are visually apparent. The authors

have quantified these lithological units by major element geochemistry, which match the visual apparent units. The 14C dating of the core is excellent. About 20% of the record appears to be in addition varve counted. The paper is well written and organized. The figures are clear. My main concern is about the data itself. The presented multiproxy data show all very similar structures, but I have to confess, that I don't see an interpretable pattern in the downcore data or time series, except those features, which are related to the apparent lithological changes. A well visible change of k-myxol at 4500 BP is the only specific change beyond those features that may be explained by the lithological units. The first prerequisite for a convincing interpretation must thus be a full documentation of the lithology. It is given as a side bar to Figs. 2, 3 and 6, but this is hardly readable. I suggest to stretch Fig. 2 on the depth scale and to document all litholological units with fotos of the sediment. This is indeed the crucial information before one can decide, if the interpretations of the many proxy curves are sound. The multiproxy time series shows the major changes in the depth interval of the section with many slumps. The slumps should be deleted from the figures on age scale. In addition the source of the lithogenic matter and its sedimentation processes should be inferred before the start of paleoenvironmental interpretations. Another clear signature is a spike of almost all organic components at about 2000 BP and in the year 1996. What happened in 1996? Was it a climatic anomaly? Was there any construction work in the catchment? The authors should make use of this historical information to "calibrate" their signals. The authors should also present the main pollen records in direct comparison to their two main organic proxies. All interpretations might become much more convincing just by an appropriate visualization. In summary, I don't feel capable of coming to a final evaluation of this manuscript. I suggest the authors add the missing information (lithology with details, fotos of sediments, pollen profiles) and provide convincing explanations for the spikes near 2000 BP and 1996 AD. It would need a new figure with only those 5 or 7 proxies, which allow a convincing synthesis. Such a synthesis figure could show a well readable lithology, two pollen demonstrating the absence of humans, two high resolution HSI and three HPLC records, all well scaled

– to indeed document the major changes - and not just many, many similar organic records. If this figures shows a clear pattern, and the signal of 1996 is understood, the study might become an excellent record from a beautiful site.

---

## Author Comment (AC1) · 22 Dec 2020

Makri et al present a very detailed record of variations in phytoplankton community composition and associated changes in redox conditions of a Polish lake. This lake has already been studied thoroughly in previous publications. However, the authors present new data together with these published records to make a very nice comparison of pigments and trace element records. Very elegant is the combination of high

resolution techniques to reconstruct short fluctuations in environmental conditions that the lake experienced with traditional techniques that provide high compound specificity, though at the expense of the high temporal resolution. The combination of these techniques provides large insight into the changes in water column conditions and species composition during the lake's history. This manuscript is suitable for Biogeosciences after consideration of mostly minor comments as outlined below.

General response: We greatly appreciate the feedback and constructive comments provided by the Anonymous Referee #1. We have addressed the comments point by point below (comments and our response right below). We agree with all the comments and we will implement all corresponding modifications in a revised version of the manuscript.

Title: redox dynamics

Line 15: altered mixing regimes – what does that mean? Is this aspect related to hypoxia or any other reasons? I guess the main problem with changed mixing regime is the change from a well-mixed system to meromixis? Please clarify.

Response: Yes, here we refer to changes pointing towards less frequent mixing in lakes. This was not clear enough. We have changed the sentence that now reads "Global spread of hypoxia and less frequent mixing in lakes is a growing major environmental concern".

Line 19: change sentence so that you state pigment analysis using two different techniques. While one method enables high spatial resolution pigment analysis (though only raw data), the HPLC data allow high compound specificity. This should be better explained here.

Response: Following the suggestion of the reviewer, we have modified the sentence so that the use of the different methods and measurement of bulk or specific compounds is clearly stated. The sentence now reads "We used a multi-proxy approach combining high-resolution bulk pigment data measured by Hyperspectral Imaging (HSI), with lower resolution specific chlorophylls and carotenoids measured by HPLC to examine Holocene trophic state changes. . .".

Line 43: The Diaz and Rosenberg papers about Hypoxia would be important references to cite here.

Response: Yes, we have added the citation here.

Line 73: Total chlorophylls or only Chlorophyll-a and derivatives considered here? Including Chlorophyll-b and c? Please clarify.

Response: Yes, thank you for this remark. We have added here more details. TChl refers mainly to Chl a and b, and their derivatives. We have also added this information in the Methods section (ref. 1st ms: p. 4, line 155).

Line 84: Related to my comment above. Please reformulate to low-resolution pigment record using HPLC analysis with high compound specificity, which cannot be achieved by the hyperspectral record.

Response: Following the suggestion of the reviewer the sentence now reads "we combined a high-resolution HSI-inferred record of TChl and Bphe, X-ray fluorescence (XRF) elemental data, and a low-resolution pigment record using HPLC analysis with high compound specificity, which cannot be achieved by the HSI record".

Line 90: Remove sentence 'This is rare in Europe.' This sentence is not useful.

Response: The sentence is removed.

Line 99: Remove Butz et al. in brackets, because it is noted twice.

Response: Corrected.

Line 157: Are bacteriopheophytin a and b both detected and distinguished by hyperspectral and HPLC techniques? It would be better to separate the records of both

compounds to establish if species-composition changes in the sedimentary record of the lake need to be considered for the reconstructions, because both compounds are not necessarily produced in the same quantities from the same species.

Response: Here we refer to total Bphe a and b. The two compounds cannot be separated using absorption spectra because their absorption almost overlaps. We have added here the word "total" so it is clear we refer to the sum of Bphe a and b and not to each compound separately.

Line 160: Are bacteriochlorophyll c, d and e present as well? If so, are they reconstructed by the HPLC technique? This also shows that the different bacteriochlorophylls and their pheophytins should be distinguished throughout the manuscript instead of using Bphe as abbreviation for the sum of these compounds.

Response: Bacteriochlorophyll c, d and e cannot be measured by HSI because their absorption overlaps with Chl a, b and their derivatives. Additionally, the extinction coefficients of these compounds are poorly constrained (or even unknown in the literature). Bphe c, d, e were absent in the HPLC record. Most probably, their concentration was very low to be detected since these compounds are very labile. We have added the wording "total Bphe a and b" for the definition and use of the word Bphe, and we trust that it is now clear that we refer to bulk Bphe a Âň+ b and not the specific compounds. We avoided using TBphe because as we explain in the text not all Bphes can be measured using the HSI RABD we refer to (ref. 1st ms: p. 4-5, lines 156-161).

Line 183: blue–green algae are also cyanobacteria. Please distinguish which forms of cyanobacteria can be reconstructed by these two pigments or are these indicators widespread in all cyanobacteria?

Response: Yes, according to Jeffrey et al. (2011) and Guilizzoni and Lami (2002), echinenone and zeaxanthin are common and most abundant in all cyanobacteria or blue-green algae. Other pigments such as myxoxanthophyll and canthaxanthin can be used to distinguish colonial and filamentous cyanobacteria since these pigments are

more abundant in these taxa. To make these more clear we have added the information in the text stating that "Echinenone and zeaxanthin are associated to most taxa of blue–green algae. . ..." .

Line 186: Pheophorbide a is considered as indicator of grazing – Please add reference to support this. It is a derivative of chlorophyll like other derivatives and can also simply form by degradation/structural alteration, which is not limited to grazing.

Response: Several authors have reported pheophorbides a to be a degradation product of Chl a transformed by microbial processes, and a useful biomarker of the effects of grazing (Bianchi and Findlay, 1991; Cartaxana et al., 2003). We have added this information and relevant references in the manuscript. The text now reads "Pheophorbide a is a degradation product of Chl a transformed by microbial processes and used as an indicator of grazing (Bianchi and Findlay, 1991; Cartaxana et al., 2003)".

Line 233: Unclear why the age uncertainty is high in the varved part of the sedimentary record. These are annual layers, so age determination should be up to a few years only? How to explain this?

Response: Thank you for this comment. As stated in the Methods section the Age-Depth model was based on 18 radiocarbon AMS dates on taxonomically identified terrestrial plant macrofossils hence the ranges of uncertainty. Establishing a varve chronology was beyond the scope of our study but is planned for the future.

Line 331: The chronology is robust and exclusively based on terrestrial macrofossils – Related comment to the previous comment. Why is there no higher precision in the age record of the upper part of the record as it is varved? Other radiometric dating techniques that are useful, such as 210Pb dating? The high uncertainty of about 140 years indicates that the age model appears less robust than it is expected to be due to the presence of varves?

Response: As mentioned above, establishing a varve chronology was beyond the

scope of our study, but this is planned in a future PhD project. The density and qual-
ity (identified terrestrial macrofossils) of our 14C ages can be considered very good,
and certainly adequate for the purpose of our study covering a period of 9500 years.
In several sections of the sediment core, varves are extremely thin; thin sections and
microscopic analysis would be required for varve counting. We do agree that varves
counting or 210Pb dating for the top part would reduce the uncertainty significantly.
Nonetheless, for the top 10 cm of the core (data shown in Fig. 4b), where HSI indices
variability was high, we used the HSI data of our core for stratigraphic correlation with
the 210Pb-dated core of Butz et al. (2016) which shows typical age errors of 3-5 years
for this part of the core (Section 4.1 in the ms).

Line 481: The data should be uploaded to PANGAEA now so the link to the datasets
can be included into the final version of the paper.

Response: The PANGAEA platform informed us that due to maintenance our data will
be uploaded with delay. Hence, we have uploaded the data to BORIS and the link is
now added in the manuscript.

References

Bianchi, T.S., Findlay, S., 1991. Decomposition of Hudson estuary macrophytes:
Photosynthetic pigment transformations and decay constants. Estuaries 14, 65–73.
https://doi.org/10.2307/1351983

Butz, C., Grosjean, M., Poraj-Górska, A., Enters, D., Tylmann, W., 2016. Sedimentary
Bacteriopheophytin a as an indicator of meromixis in varved lake sediments of Lake
Jaczno, north-east Poland, CE 1891–2010. Glob. Planet. Change 144, 109–118.
https://doi.org/10.1016/j.gloplacha.2016.07.012

Cartaxana, P., Jesus, B., Brotas, V., 2003. Pheophorbide and pheophytin a-like pig-
ments as useful markers for intertidal microphytobenthos grazing by Hydrobia ulvae.
Estuar. Coast. Shelf Sci. 58, 293–297. https://doi.org/https://doi.org/10.1016/S0272-

7714(03)00081-7

Guilizzoni, P., Lami, A., 2002. Paleolimnology: Use of Algal Pigments as Indicators, in: Bitton, G. (Ed.), Encyclopedia of Environmental Microbiology. John Wiley & Sons, Inc., pp. 2306–2317. https://doi.org/https://doi.org/10.1002/0471263397.env313

Jeffrey, S., Wright, S., Zapata, M., 2011. Phytoplankton Pigments, Phytoplankton Pigments. https://doi.org/10.1017/cbo9780511732263

---

## Author Comment (AC2) · 22 Dec 2020

The paper by Makri et al makes use of a high resolution, laminated lake sediment record from Poland, which covers the last 9500 years. The authors use high-resolution (mm-scale) Hyperspectral Imaging pigment data together with low resolution (dm scale) chlorophyll and caretonoids data to document the impact of humans into the lake and nearby environment. The lake is particularly suited for such a study, because

pollen evidence document that the region is used by humans only since about 500 years. The region was in a natural state apparently for most of the Holocene. The lithology is presented as three main units, which are visually apparent. The authors have quantified these lithological units by major element geochemistry, which match the visual apparent units. The 14C dating of the core is excellent. About 20% of the record appears to be in addition varve counted. The paper is well written and organized. The figures are clear.

General response: We greatly appreciate the careful revision and the constructive comments provided by the Anonymous Referee #2. We have addressed the concerns each by each below. We give our response right below each comment. We understand and we mostly agree with the concerns of the Reviewer and we trust that our responses and subsequent modifications in a revised manuscript will clarify and sharpen our interpretation and focus of our paper.

Comment 1: My main concern is about the data itself. The presented multiproxy data show all very similar structures, but I have to confess, that I don't see an interpretable pattern in the downcore data or time series, except those features, which are related to the apparent lithological changes. A well visible change of k-myxol at 4500 BP is the only specific change beyond those features that may be explained by the lithological units. The first prerequisite for a convincing interpretation must thus be a full documentation of the lithology. It is given as a side bar to Figs. 2, 3 and 6, but this is hardly readable. I suggest to stretch Fig. 2 on the depth scale and to document all lithological units with fotos of the sediment. This is indeed the crucial information before one can decide, if the interpretations of the many proxy curves are sound.

Response: Climate and catchment evolution changes are the main drivers of pigment variability in Lake Jaczno, as explained in the text. Indeed, these changes are also registered and reflected in the lithology. Thus, they coincide largely with the lithological units, which is very interesting here. It is not always the case that lithological units and pigment or other organic proxies are so consistent. For Figs. 2, 3 and 6 we have en-
larged the column with the lithological units so they are now better readable and clear. The detailed documentation of all lithological units asked by the reviewer is provided in the supplementary material (Fig. S3). This figure shows all lithological units with photo documentation of sediment structures. In our opinion, this is the appropriate place and avoids an overly long main text. A clear reference of the content and presence of this Figure is found at the beginning of Section 4.3.

Comment 2: The multiproxy time series shows the major changes in the depth interval of the section with many slumps. The slumps should be deleted from the figures on age scale. In addition the source of the lithogenic matter and its sedimentation processes should be inferred before the start of paleoenvironmental interpretations.

Response: Indeed, the slumps were removed prior to the chronological modeling (as explained in the text) but we decide to leave them in the Figure. This information might be helpful for other research groups working in this lake in the future (helps the stratigraphic correlation of cores). The sources of the lithogenic sediments are in the catchment. According to our interpretation model (see text Section 5.2) the lithogenic components are indicative of surface processes in the catchment (erosion).

Comment 3: Another clear signature is a spike of almost all organic components at about 2000 BP and in the year 1996. What happened in 1996? Was it a climatic anomaly? Was there any construction work in the catchment? The authors should make use of this historical information to "calibrate" their signals.

Response: We attribute these peaks to warmer summer temperatures; this is, however, especially for the period around 2000 cal BP, not well established. Nonetheless, 1996 CE was quite unusual in terms of climatic conditions. The winter was very long and the temperatures were low until around April 20th (Czernecki and MiÄŹtus, 2017). Since April 20, temperatures increased very quickly together with very warm airflow from North Africa. Temperatures reached 25-27 °C during the day and even 10-15 °C at night ("IMGW-PIB, Suwałki Meteorological Station," 2017). Hence, after a long
winter with thick ice cover, summer stratification developed almost immediately. These climate conditions were very similar to 2013 CE (long and cold winter combined with hot spring and long summer stratification, Fig. 1d). Also 1997 was similar: the winter was long (until April) and then a very warm spring. Increased summer temperature registered after 1990 CE and eutrophication (maximal pigment concentrations) had a positive effect on the persistence of meromixis (Butz et al., 2016). In our interpretation, we place emphasis on the long-term trends (not individual data points) as shown in the Zones I – IV of the RDA (Fig 5); Fig. 5 shows the differences between the pigment zones and their relation with temperature, vegetation cover and surface processes. We have added this information about the warming of the temperature after 1990 CE in the text in Section 5.2.3.

Comment 4: The authors should also present the main pollen records in direct comparison to their two main organic proxies. All interpretations might become much more convincing just by an appropriate visualization.

Response: This is basically shown in our synthesis Fig. 6 displaying AP/NAP, Bphe and TChl. All details of the pollen profile have been published in Kinder et al. (2019) and Marcisz et al. (2020). From other works in lakes from Poland, Greece and Switzerland, we know that it is mostly the AP/NAP pollen ration (or the density of the forest) that influences the mixing regime (i.e. which is the purpose of our paper).

Comment 5: In summary, I don't feel capable of coming to a final evaluation of this manuscript. I suggest the authors add the missing information (lithology with details, fotos of sediments, pollen profiles) and provide convincing explanations for the spikes near 2000 BP and 1996 AD. It would need a new figure with only those 5 or 7 proxies, which allow a convincing synthesis. Such a synthesis figure could show a well readable lithology, two pollen demonstrating the absence of humans, two high resolution HSI and three HPLC records, all well scaled – to indeed document the major changes - and not just many, many similar organic records. If this figures shows a clear pattern, and the signal of 1996 is understood, the study might become an excellent record from
a beautiful site.

Response: As mentioned above, the information about the lithology (with pictures) is shown in the supplementary material (Fig. S3). Also the most important proxies supporting the arguments (and purpose) of our paper are already shown in our synthesis figure Fig. 6 (pollen, temperature, both hyperspectral indices, the most diagnostic pigments indicating anoxia, and lithogenic flux).

References

Butz, C., Grosjean, M., Poraj-Górska, A., Enters, D., Tylmann, W., 2016. Sedimentary Bacteriopheophytin a as an indicator of meromixis in varved lake sediments of Lake Jaczno, north-east Poland, CE 1891–2010. Glob. Planet. Change 144, 109–118. https://doi.org/10.1016/j.gloplacha.2016.07.012

Czernecki, B., MiÄŹtus, M., 2017. The thermal seasons variability in Poland, 1951–2010. Theor. Appl. Climatol. 127, 481–493. https://doi.org/10.1007/s00704-015-1647-z

IMGW-PIB, Suwałki Meteorological Station [WWW Document], 2017. . Inst. Meteorol. Water Manag. - Natl. Res. Inst. URL https://www.imgw.pl/

Kinder, M., Tylmann, W., Bubak, I., Fiłoc, M., GÄĚsiorowski, M., Kupryjanowicz, M., Mayr, C., Sauer, L., Voellering, U., Zolitschka, B., 2019. Holocene history of human impacts inferred from annually laminated sediments in Lake Szurpiły, northeast Poland. J. Paleolimnol. 61, 419–435. https://doi.org/10.1007/s10933-019-00068-2

Marcisz, K., Kołaczek, P., Gałka, M., Diaconu, A.-C., Lamentowicz, M., 2020. Exceptional hydrological stability of a Sphagnum-dominated peatland over the late Holocene. Quat. Sci. Rev. 231, 106180. https://doi.org/https://doi.org/10.1016/j.quascirev.2020.106180

Please also note the supplement to this comment:

BGD
https://bg.copernicus.org/preprints/bg-2020-362/bg-2020-362-AC2-supplement.pdf

---

## Author Response (AR2)

Manuscript ID bg-2020-362 titled "Holocene phototrophic community and anoxia dynamics in meromictic Lake Jaczno (NE Poland) using high-resolution hyperspectral imaging and HPLC data".

Comments to the Author:

Dear Dr. Makri,

It is with great pleasure that I read your revision and I am happy to recommend it for publication in BG provided that you fix a few small issues that I noticed. Overall, may I encourage you to double-check the text for typos. Please see for instance line 293 (In zone III (6700–500 cal BP), most pigments concentration increase gradually.). Furthermore, there are some problems with the references (lines 524 and 557 etc) likely due to the conversion from the citation manager.

Thank you very much for fixing these. Looking very much forward to seeing your work published in BG.

My very best regards,

Tom Battin

Response:

We would like to thank the Editor for these last remarks and helpful comments to improve the quality of our manuscript. We have corrected the grammatical errors in line 293, the references content in lines 524 and 557, and we have doubled checked the entire manuscript for typos and further editing errors. Below we attach the manuscript with track changes where all last corrections are visible.

[revised manuscript text omitted]

- $763 \qquad \text{radiocarbon measurements} \ (